# Determinants of return to HIV treatment after interruption on ART among HIV positive clients in Katakwi District, Uganda

**William Okello**[1]*, **Saadick Mugerwa Ssentongo**[1], **Bonniface Oryokot**[1], **Baker Bakashaba**[1], **Ronald Opito**[1,2], **Bosco Opio**[3], **Christine Acio**[3]

1 Directorate of Program Management and Capacity Development, AIDS Information Center (AIC), Kampala, Uganda, 2 Department of Public Health, School of Health Sciences, Soroti University, Soroti, Uganda, 3 Department of Public Health, School of Health Sciences, Lira University, Lira, Uganda

* okellowilliam877@gmail.com

## Abstract

### Background

Returning to treatment following interruptions is crucial for achieving optimal HIV care outcomes. In Uganda, despite a 20% treatment interruption rate, only 58% of clients successfully resume treatment. Evidence on determinants of returning to treatment remains limited. This study aimed to identify determinants of return to HIV treatment after interruption in Katakwi District, Uganda.

### Methods

We conducted a cross-sectional study at three high-volume antiretroviral therapy (ART) clinics in Katakwi District. Data were collected using face-to-face interviews from study adult participants and analyzed using Poisson generalized estimating equations (GEE) with robust standard errors to identify factors independently associated with a return to care.

### Results

The rate of return to care was 63.9%. Clients without an occupation were less likely to return (aRR = 0.80, 95% CI: 0.73–0.88, p < 0.001). Surprisingly, those living within 5 km of a facility were also less likely to return compared to those 5–10 km away (aRR = 0.78, 95% CI: 0.63–0.95, p = 0.019). Positive community beliefs about treatment adherence significantly increased the likelihood of returning (aRR = 1.18, 95% CI: 1.06–1.33, p = 0.003), as did belonging to a community support group (aRR = 1.16, 95% CI: 1.01–1.39, p = 0.04).

**Data availability statement:** All relevant data are within the paper and its Supporting Information files.

**Funding:** The author(s) received no specific funding for this work.

**Competing interests:** The authors have declared that no competing interests exist.

## Conclusion

Return to HIV care was associated with occupation, distance to the facility, community beliefs, and belonging to a community support group. Interventions to improve re-engagement should include targeted economic support for unemployed clients, community-based anti-stigma campaigns, flexible clinic hours, and improved rural access through mobile clinics. Continuous client education at both the facility and community levels is essential.

## Introduction

Return to treatment refers to clients' initiative to reinitiate care after a gap of 28 days or more from the last expected clinical contact or drug pick up [1]. Globally, despite the availability of safe ART, only 28.2 million out of the 37.7 million people living with HIV (PLHIV) are currently accessing antiretroviral therapy (ART), a situation resulting from poor retention in care stemming from a sub-optimal return to treatment [1–3]. Global estimates of return to treatment rates remain low, varying from 32.7% in America, 12.1% in Europe, and around 39.4% in Africa [4]. In sub-Saharan Africa, which has an estimated 16 million people living with HIV, only 12 million are currently in treatment, despite tremendous scale-up activities to increase access to ART [5–7].

In Uganda, 80.9% of adults living with HIV were aware of their HIV status, and of these, 96.1% were on ART [8]. Although the country has made commendable strides in increasing ART coverage, treatment interruption rates are as high as 12%, well above the acceptable threshold of 2%. In response, Uganda has implemented various interventions to improve return to treatment, including using physical locator forms, phone calls, and physical tracing of clients who interrupt treatment. Despite these efforts, retention rates are still suboptimal at 81%, far below the Joint United Nations Program on HIV/AIDS (UNAIDS) target of 95%, and also return to treatment (RTT) of only 58%, short of the 95% UNAIDS target [9,10].

Katakwi district has one of the highest numbers of clients in care (5,600) in the Teso Sub-region, second only to Soroti City. It registered a high treatment Interruption rate of 4% with a low RTT rate of only 67% compared to the rest of the districts in the region in January-March 2024 [11]. The dynamic movement of patients in and out of HIV care is prevalent, but there is limited information on return to treatment predictors to guide HIV programs to better support patient engagement [12,13].

Several studies have identified various reasons why people interrupt their HIV treatment, including individual factors such as fear, unstable income, forgetting to take medication, and negative attitudes towards healthcare workers, stigma, lack of social support, poor community support systems, and unsupportive clinic structures, as well as other associated factors that may contribute to a return to care [2,10,14–17]. Interruption in treatment results in clinical deterioration, viral resistance, persistent viremia, ongoing HIV transmission, poor health outcomes, high mortality, and loss of economic opportunities. Up to 30% of hospital HIV-related admissions occur amongst people who interrupt treatment [10,12,18–20]. This study aimed to examine

determinants of return to HIV treatment after interruption in Katakwi District, Uganda. The findings will inform the development of strategies to re-engage lost patients, ensure treatment adherence, achieve viral load suppression for improved health outcomes.

## Materials and methods

### Study design

This was a cross-sectional study. Data were collected using a structured questionnaire by 31st July 2024. The findings were reported following the Strengthening the Reporting of Observational Studies in Epidemiology (STROBE) guidelines [21].

### Study population

The Study population consisted of PLHIV aged 18 years and above with a history of treatment interruption who either returned to care or had not yet successfully returned to care. Participants were recruited from three ART sites in Katakwi District, purposively selected for having the highest client volumes in the district.

### Study setting

The study was conducted at three high-volume ART clinics in Katakwi District: Katakwi General Hospital, Magoro Health Center III, and Ngariam Health Center III, which together serve 73% of the total PLHIV in the district. Katakwi District had a total of 5,600 PLHIV as of January-March 2022 and is located approximately 350 km northeast of Kampala. The district has a total population of 234,705 according to the 2024 Uganda National Housing and Population study [22]. The district is characterized by a semi-arid climate, with cattle-keeping and subsistence farming as the major economic activities. The district borders Karamoja to the east, and insecurity caused by cattle rustlers may hinder return to care [23].

### Inclusion and exclusion criteria

We included all PLHIV 18 years and above receiving care from the three high-volume sites, who had experienced treatment interruption between October 2023 and June 2024. We excluded clients with a history of psychiatric illness impairing their ability to provide informed consent, those who did not consent to the study, incarcerated individuals, those too ill or admitted, and those residing outside the Katakwi District.

### Sample size estimation

A sample size of 355 was calculated using the Kish-Leslie formula (1965) [24] for a cross-sectional study in a single population with an estimated return to treatment following interruption at 70% from a previous study [25], a type 1 error of 5%, a 95% confidence level (Z = 1.96), and a 10% non-response rate. The sample size was then distributed proportionally among three selected facilities based on their PLHIV caseload in care as of the January-March 2024: 229 from Katakwi General Hospital, 65 from Magoro Health Center III, and 61 from Ngariam Health Center III.

### Sampling procedures

Participants with a history of treatment interruption were identified through the Open Electronic Medical Record System (EMRS) at the three sites. The identified participants were cross-checked with their paper-based charts in real time to ensure data accuracy and minimize misclassifications due to missing or incorrect data entries. A sampling frame was created from this list, and participants were selected using consecutive sampling. All eligible clients with appointments during the study period were interviewed; those with later appointments were called back for an interview. Clients who had interrupted treatment but not returned were followed up and interviewed.

## Study variables

**The Primary Outcome** was Return to treatment, defined as reinitiating treatment after a gap in care of more than 28 days [26].

Interruption in treatment (IIT), defined as no clinical contact for 28 days or more after the last expected clinical contact or drug pick up [22].

**Lost to Follow Up** (LTFU) was defined as no clinical contact for at least 90 days after the last expected clinical visit [27].

**The Independent Variables** included socio-demographic characteristics (age, Sex, educational level, occupation, and marital status), clinical factors (duration on ART, Mode of care, Line of treatment (first, second, and third line), Distance to clinic, HIV status Disclosure, community factors (belonging to community support groups, and beliefs in the community against ITT and community networks) and Institutional factors (Clinic Operating Hours, adequacy of Privacy, having received Phone/Short Message Service (SMS) reminders, experience of Drug Stock Out, and adequacy of Health workers.

**Data collection.** Data on clients who had missed appointments was first collected by trained research assistants from the Open EMRS system and triangulated with paper-based charts and registers to ensure accuracy and completeness, and to avoid misclassification. From this extracted data, individuals with a history of treatment interruption (both those who had returned to care and those who had not yet returned to care) were identified. We then further collected data from these identified individuals through face-to-face interviews administered by structured questionnaires. The interview lasted between 10–20 minutes using a questionnaire based on the existing literature and translated into the local language (Ateso).

**Data analysis.** The data were entered into Microsoft Excel, where it was cross-checked and validated for correctness and completeness. After checking for missing data and unsound entries, the data were exported to STATA version 17 (College Station, TX: StataCorp LLC) [28]. Descriptive statistics were summarized using proportions for categorical variables, means with SDs for continuous variables with normal distributions, and medians with interquartile ranges (IQRs) for continuous variables with skewed distributions. The association of return to treatment and categorical variables was assessed using the Chi-square test or Fisher's exact tests, and associations between return to treatment and numerical variables were assessed using Student's t-test or Mann–Whitney U tests. We used Poisson generalized estimating equations (GEE) with robust standard errors, accounting for clustering at the health facility level and adjusting for Variables with a $p < 0.2$ at bivariate analysis and clinically relevant variables identified in the literature. The results were presented as Crude Risk ratios (cRRs) and adjusted Risk Ratios (aRRs) with a 95% confidence interval. A p value of $< 0.05$ was considered statistically significant for analysis.

**Ethical approval and consent to participate.** The study received ethical approval from Lira University Research and Ethic Committee (LUREC-2024–235). Participants provided written informed consent, and each participant retained a signed copy after the study. Administrative clearance and permission were obtained from the District Health Officer of Katakwi District, the Hospital Medical Superintendent of Katakwi Hospital, and the person in charge of Magoro and Ngariam H/C III. Confidentiality was maintained by ensuring that individual patient-level data obtained was de-identified, encrypted, and passworded to ensure access by only an authorized team of investigators.

## Results

### Recruitment

Of the 386 clients with a history of treatment interruption who were approached, 20 declined participation, 5 were misclassified as having interrupted treatment, yet they had picked up their ART, and 6 had transferred to other facilities. A total of 355 clients were included in the final analysis (Fig 1).

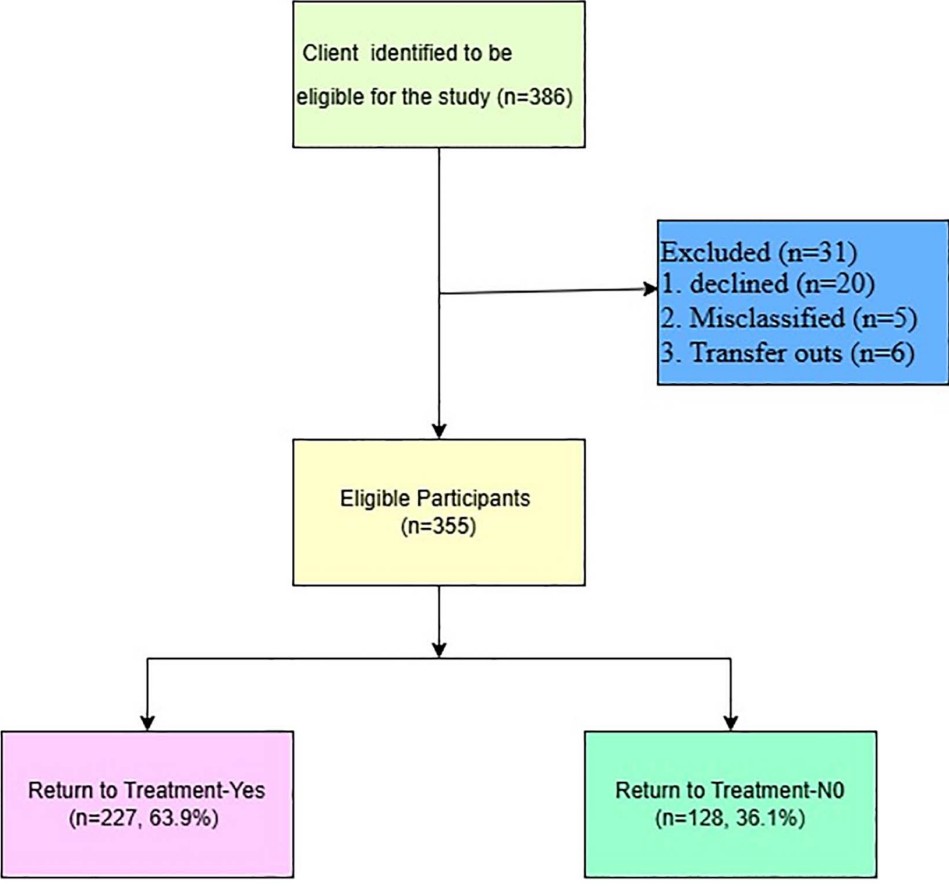

**Fig 1. Study flowchart for the determinants of return to HIV treatment after interruption on ART among HIV positive clients in Katakwi District.**

## Baseline characteristics

In this study, the majority of the participants were aged 18–30 years (67.6%, n = 240) with a median age of 44 (interquartile range of 35–52). The majority (57.2%, n = 203) were female. In terms of education, 52.1% (n = 185) had primary education, 27.9% (n = 99) had no formal education, 13.2% (n = 47) had secondary education, and 6.8% (n = 24) had tertiary education. More than three-quarters of respondents had an occupation (93.8%, n = 333). In this study, 65.4% (n = 232) of the participants were married, 22.8% (n = 81) were single, and 11.8% (n = 42) were widowed as shown in Table 1 below.

## Prevalence of return to treatment following interruption in treatment

Among the PLHIV, 63.9% (n = 227, CI: 58.8%–68.7%) returned to treatment after an interruption, while 36.1% (n = 128) did not.

## Determinants of return to HIV treatment after interruption

In bivariate analysis, mode of care, distance to the clinic, community beliefs about treatment interruption, and adequate privacy during service provision were significantly associated with returning to care.

Clients in the community care model had a higher return rate (83.9%) compared to those in the facility model (57.4%). Furthermore, clients living within 5 km of the clinic were less likely to return to treatment (50.4%) compared to those living

**Table 1. Social demographic characteristics.**

| Variables | Frequency, N = 355 (%) | |
|---|---|---|
| Median Age: 44 (interquartile range: 36–52) | | |
| **Age category in years** | | |
| 18-30 | 240 | 67.6 |
| 31-40 | 70 | 19.7 |
| 41-50 | 30 | 8.5 |
| 50+ | 15 | 4.2 |
| **Gender** | | |
| Female | 203 | 57.2 |
| Male | 152 | 42.8 |
| **Education level** | | |
| None | 99 | 27.9 |
| Primary | 185 | 52.1 |
| Secondary | 47 | 13.2 |
| Tertiary | 24 | 6.8 |
| **Occupation** | | |
| No | 22 | 6.2 |
| Yes | 333 | 93.8 |
| **Marital Status** | | |
| Married | 232 | 65.4 |
| Single | 81 | 22.8 |
| Widowed | 42 | 11.8 |
| **Duration on ART** | | |
| ≤ 5 years | 73 | 20.6 |
| > 5years | 282 | 79.4 |
| **Mode of care** | | |
| Community Model | 87 | 24.5 |
| Facility Model | 268 | 75.5 |
| **Treatment line** | | |
| First Line | 325 | 91.5 |
| Second Line | 24 | 6.8 |
| Third Line | 6 | 1.7 |
| **Distance to clinic** | | |
| < 5 kms | 121 | 34.1 |
| 5–10 kms | 139 | 39.2 |
| > 10 kms | 95 | 26.8 |
| **Disclosed HIV status** | | |
| No | 22 | 6.2 |
| Yes | 333 | 93.8 |
| **Belonging to a community Support Group** | | |
| No | 116 | 32.7 |
| Yes | 239 | 67.3 |
| **Beliefs against IIT in the community** | | |
| **No** | 186 | 52.4 |
| Yes | 169 | 47.6 |
| **Community Support networks for follow-up** | | |
| No | 42 | 11.8 |

*(Continued)*

**Table 1.** (Continued)

| Variables | Frequency, N = 355 (%) | |
|---|---|---|
| Yes | 313 | 88.2 |
| **Favourable Clinic Operating Hours** | | |
| No | 49 | 13.8 |
| Yes | 306 | 86.2 |
| **Adequate Privacy at the facility** | | |
| No | 83 | 23.4 |
| Yes | 272 | 76.6 |
| **Received Reminder calls/SMS for appointment** | | |
| No | 102 | 28.7 |
| Yes | 253 | 71.3 |
| **Drug stockouts during refills** | | |
| No | 222 | 62.5 |
| Yes | 133 | 37.5 |
| **Availability of enough health workers to offer services.** | | |
| No | 146 | 41.1 |
| Yes | 209 | 58.9 |

5–10 km away (69.1%) and those living more than 10 km away (73.7%). Clients in communities with beliefs discouraging treatment interruption had a higher rate of return to care (79.3%) compared to those in communities without such beliefs (50.0%). Clients who experienced adequate privacy were more likely to return to treatment (66.9%) compared to those who did not (54.2%), as shown in Table 2.

In the adjusted multivariate analysis as shown Table 3 below, the factors associated with returning to treatment following an interruption among HIV-positive clients in Katakwi District included having no occupation, living within 5 km of the facility, living in a Community with beliefs against interruption of treatment and Belonging to the community support groups. PLHIV with no occupation were 20% less likely to return to treatment compared to those with an occupation (aRR = 0.80, 95% CI: 0.73–0.88). Those living within 5 km of the facility were 22% less likely to return to treatment than those living 5−10 km away (aRR = 0.78, 95% CI: 0.63–0.95, p = 0.019), living in a Community with beliefs against interruption of treatment was strongly associated with a higher likelihood of returning to treatment (aRR = 1.18, 95% CI: 1.06–1.33, p = 0.003). Similarly, belonging to the community support group was aaociated with a higher likelihood (aRR 1.16, 95% CI: 1.01–1.39, p = 0.04).

## Discussion

In this study, we found that 63.9% of HIV-positive clients in Katakwi District returned to treatment after an interruption. This rate of return is notably higher than some of the findings in other studies done in Uganda and South Africa that reported a return to treatment rate of 42% to 70% [9,10,25,29,30], with higher rates typically seen in rural contexts. The higher return rate observed in the study is attributed to localized factors, such as community support, quality improvement initiatives, and active bringing back to care activities by the district health teams and the local implementing partner. Research on return to care has noted disparities with higher rates of up to 70% in rural areas as compared to urban areas, which is particularly due to stronger client navigation support systems and flexibility in these settings [10,1231].

We found that clients without occupation were less likely to return to HIV treatment than employed individuals. This is consistent with studies done in South Africa and Ethiopia [13,32]. This is because poor economic status is associated with a lack of transport to the clinic and the inability to take medication due to lack of food, among other factors, resulting in interruption in treatment [29,32].

**Table 2. Determinants of return to HIV treatment after interruption in Katakwi District, Uganda.**

| Variables | N = 355 | Return to treatment | | p-value |
|---|---|---|---|---|
| | | No (n = 128, 36.1%) | Yes (n = 227, 63.9%) | |
| **Age category in years** | | | | |
| 18-30 | 240 (67.6) | 86 (35.8) | 154 (64.2) | 0.971 |
| 31-40 | 70 (19.7) | 26 (37.1) | 44 (62.9) | |
| 41-50 | 30 (8.5) | 10 (33.3) | 20 (66.7) | |
| 51+ | 15 (4.2) | 6 (40.0) | 9 (60.0) | |
| **Gender** | | | | |
| Female | 203 (57.2) | 72 (35.5) | 131 (64.5) | 0.79 |
| Male | 152 (42.8) | 56 (36.8) | 96 (63.2) | |
| **Education level** | | | | |
| None | 99 (27.9) | 38 (38.4) | 61 (61.6) | 0.655 |
| Primary | 185 (52.1) | 66 (35.7.0) | 119 (64.3) | |
| Secondary | 47 (13.2) | 18 (38.3) | 29 (61.7) | |
| Tertiary | 24 (6.8) | 6 (25) | 18 (75) | |
| **Occupation** | | | | |
| No | 22(6.2) | 11 (50.0) | 11 (50.0) | 0.16 |
| Yes | 333 (93.8) | 117 (35.1) | 216 (64.9) | |
| **Marital Status** | | | | |
| Married | 232 (65.4) | 82 (35.3) | 150 (64.7) | 0.219 |
| Single | 81 (22.8) | 26 (32.1) | 55 (67.9) | |
| Widowed | 42 (11.3) | 20 (47.6) | 22 (52.4) | |
| **Duration on ART** | | | | |
| ≤5 years | 73 (20.6) | 24 (32.9) | 49 (67.1) | 0.526 |
| >5years | 282 (79.4) | 104 (36.9) | 178 (63.1) | |
| **Mode of care** | | | | |
| Community Model | 87 (24.5) | 14 (16.1) | 73 (83.9) | **0.001*** |
| Facility Model | 268 (75.5) | 114 (42.5) | 154 (57.4) | |
| **Treatment line** | | | | |
| First Line | 325 (91.5) | 119 (36.6.0) | 206 (63.4) | 0.402 |
| Second Line | 24 (6.8) | 6 (25.0) | 18 (75.0) | |
| Third Line | 6 (1.7) | 3 (50.0) | 3 (50.0) | |
| **Distance to clinic** | | | | |
| <5 kms | 121 (34.1) | 60 (49.6) | 61 (50.4) | |
| 5–10 kms | 139 (39.2) | 43 (30.9) | 96 (69.1) | **0.001*** |
| >10 kms | 95 (26.8) | 25 (26.3) | 70 (73.7) | |
| **Disclosed HIV status** | | | | |
| No | 22 (6.2) | 5 (22.7) | 17 (77.3) | 0.179 |
| Yes | 333 (93.8) | 123 (36.9) | 210 (63.1) | |
| **Belonging to a community Support group** | | | | |
| No | 116 (32.7) | 49 (42.2) | 67 (57.8) | 0.091 |
| Yes | 236 (67.3) | 79 (33.1) | 160 (66.9) | |
| **Beliefs against IIT in the community** | | | | |
| **No** | 186 (52.4) | 93 (50.0) | 93 (50.0) | **0.001*** |
| Yes | 169 (47.6) | 35 (20.7) | 134(79.3) | |
| **Community Support networks for follow-up** | | | | |
| No | 42 (11.8) | 13 (31.0) | 29 (69.0) | 0.463 |

*(Continued)*

**Table 2.** (Continued)

| Variables | N = 355 | Return to treatment | | p-value |
|---|---|---|---|---|
| | | No (n = 128, 36.1%) | Yes (n = 227, 63.9%) | |
| Yes | 313 (88.2) | 115(36.7) | 198(63.3) | |
| **Favourable Clinic Operating hours** | | | | |
| No | 49 (13.8) | 20 (40.8) | 29 (59.2) | 0.455 |
| Yes | 306 (86.2) | 108(35.3) | 198(64.7) | |
| **Adequate Privacy at the facility** | | | | |
| No | 83 (23.4) | 38 (45.8) | 45 (54.2) | **0.035*** |
| Yes | 272 (76.6) | 90 (33.1) | 182(66.9) | |
| **Received Reminder calls/SMS for appointment** | | | | |
| No | 102 (28.7) | 37 (36.3) | 65 (63.7) | 0.957 |
| Yes | 253 (71.3) | 91 (36) | 162(64) | |
| **Drug stockouts during refills** | | | | |
| No | 222 62.5) | 85 (38.3) | 137(61.7) | 0.258 |
| Yes | 133 (37.5) | 43 (32.3) | 90 (67.7) | |
| **Availability of enough health worker facilities to offer services.** | | | | |
| No | 146 (41.1) | 60 (41.1) | 86 (58.9) | 0.098 |
| Yes | 209 (58.9) | 68 (32.5) | 141(67.5) | |

Note: * indicate a significant variable at 5%, km = kilometer, SMS = Short Message Service, IIT = interruption in treatment.

We also found that PLHIV living within 5 kilometers of the facility were less likely to return to treatment, a finding that contrasts with the general understanding that proximity usually supports better adherence [33]. This paradox suggests that in this context, psychosocial barriers, such as increased stigma or reduced motivation, may be more influential than distance. The role of Stigma has as a consistent barrier to treatment re-engagement is well established in the literature [34–36].

Clients from communities with strong beliefs discouraging treatment interruption were more likely to return to treatment than those from communities without such beliefs, a finding consistent with studies in the United States of America and Tanzania [37–39]. This is because positive community beliefs reduce HIV related stigma, create peer pressure for ART adherence, and foster an enabling environment for PLHIV to stay in care [40].

Belonging to community support groups was associated with higher rates of returning to care, aligning with the literature, which emphasizes the importance of social support, interpersonal relationships, and community networks in improving treatment adherence and re-engagement in care [3,4,20,41,42].

## Limitations and strengths

Due to the cross-sectional nature of this study, we can identify associations but cannot establish causality or temporal sequence between factors and return to care. We did not include facilities with varying levels of services, which may limit the generalizability of our findings to similar settings, and we did not account for a design effect. However, key strengths of the study include its multi-center design and its focus on a population that has already experienced interruption of treatment, providing direct insights into the re-engagement process.

## Conclusions

In this study, a high return-to-care rate was positively associated with participation in a community support group and residence in a community that values treatment adherence, but negatively associated with living <5 km from the

**Table 3.  Multivariable analysis of determinants of return to HIV treatment after interruption in Katakwi District, Uganda.**

| Variable | cRR (95% CI) | p-value | aRR (95% CI) | p-value |
|---|---|---|---|---|
| **Gender** | | | | |
| Male | **Reference** | | | |
| Female | 1.06 (0.97-1.15) | 0.162 | 1.03 (0.89-1.18) | 0.697 |
| **Age categories** | | | | |
| 50+ | **Reference** | | | |
| 18-30 | 1.08 (0.91-1.28) | 0.362 | 1.09 (0.73-1.64) | 0.651 |
| 31-40 | 1.11 (1.00-1.23) | 0.05 | 1.12 (0.95-1.32) | 0.164 |
| 41-50 | 1.08 (0.88-1.33) | 0.433 | 1.06 (0.87-1.30) | 0.554 |
| **Occupation** | | | | |
| Yes | **Reference** | | | |
| No | 1.25 (1.19-1.30) | <0.001* | 0.80 (0.73-0.88) | **<0.001*** |
| **Mode of care** | | | | |
| Community model | **Reference** | | | |
| Facility Model | 0.98 (0.73-1.32) | <0.001* | 1.08 (0.77-1.51) | 0.641 |
| **Distance to facility** | | | | |
| 5–10 kms | **Reference** | | | |
| <5 kms | 0.78(0.65-0.93) | 0.007* | 0.78 (0.63-0.95) | **0.019*** |
| >10 kms | 1.85 (0.64-1.13) | 0.227 | 0.87 (0.66-1.14) | 0.328 |
| **Disclosure Status** | | | | |
| No | **Reference** | | | |
| Yes | 0.92 (0.81-1.04) | 0.189 | 0.90 (0.71-1.12) | 0.362 |
| **Beliefs in the community against interruption in Treatment** | | | | |
| No | **Reference** | | | |
| Yes | 1.18 (1.08-1.30) | <0.001* | 1.18 (1.06-1.33) | **0.003*** |
| **Adequate privacy at the facility** | | | | |
| No | **Reference** | | | . |
| Yes | 0.98 (0.97-0.99) | 0.036* | 1.01 (0.89-1.13) | 0.927 |
| **Availability of enough health workers to offer services.** | | | | |
| No | **Reference** | | | |
| Yes | 0.98 (0.723-1.33) | 0.099 | 1.01 (0.75-1.34) | 0.995 |
| **Belonging to a community Support group** | | | | |
| No | **Reference** | | | |
| Yes | 1.19 (1.02-1.39) | 0.026* | 1.16 (1.01-1.39) | **0.04*** |

Note: * indicate significant variable at 5%, km = kilometer, cRR = Crude Risk Ratio, aRR = Adjusted Risk ratio.

facility and being unemployed. These findings highlight patient and health system-related gaps that require tailored interventions to ensure that PLHIV who interrupt treatment return to care for improved health outcomes. We recommend that government health agencies and non-governmental organizations (NGOs) design targeted programs, including economic empowerment initiatives like small business grants and vocational training activities for unemployed PLHIV and strengthened differentiated service delivery models, such as community-based drug distribution and flexible mobile clinics to reduce travel burdens for clients living near facilities. Finally, it's crucial to scale up community education to combat stigma, alongside robust counseling and peer support programs, to foster a supportive environment for clients struggling to return to care.

## Supporting information

**S1 Data. Data for Submissionv1.**
(CSV)

**S1 File. IIT ethical approval letter.**
(DOCX)

**S2 File. STROBE-checklist-return to care.**
(DOC)

## Acknowledgments

The authors would like to acknowledge the support rendered by the facility in charge of the Katakwi General Hospital, Magoro Health center III and Ngariam Health center III during the process of data collection for this study. In addition, the author acknowledges the contribution of study participants for their consent and acceptance to participate in this study.

## Author contributions

**Conceptualization:** William Okello, Saadick Mugerwa Ssentongo, Bonniface Oryokot, Ronald Opito, Christine Acio, Baker Bakashaba.

**Data curation:** William Okello, Saadick Mugerwa Ssentongo.

**Formal analysis:** William Okello, Bonniface Oryokot, Bosco Opio.

**Methodology:** William Okello, Saadick Mugerwa Ssentongo, Ronald Opito, Bosco Opio.

**Resources:** William Okello.

**Software:** Bonniface Oryokot.

**Supervision:** William Okello, Bonniface Oryokot, Christine Acio.

**Validation:** William Okello, Ronald Opito, Christine Acio, Baker Bakashaba.

**Visualization:** Christine Acio, Baker Bakashaba.

**Writing – original draft:** William Okello.

**Writing – review & editing:** Bonniface Oryokot, Christine Acio, Baker Bakashaba.

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
