## [Decision Letter · Decision Letter 0]

12 Sep 2025

Dear Dr. Okello,

Thank you for submitting your manuscript to PLOS ONE. After careful consideration, we feel that it has merit but does not fully meet PLOS ONE’s publication criteria as it currently stands. Therefore, we invite you to submit a revised version of the manuscript that addresses the points raised during the review process.

Please address these comments including all the comments raised by reviewers 1 and 2.

**Ethical statement:** “Ethical approval and consent to participate. The research protocol was submitted to Lira University's Faculty of Public Health and the Lira University Research and Ethic Committee for approval (LUREC-2024-235)”.

Comment: the statement is unclear whether the study has been approved by the ethics committee or otherwise. Please be explicit about this.

**Introduction:**

Line 54: In Uganda, out of an estimated 1,492,742 people living with HIV, 1,255,975 are on ARVs (11). Comment: please provide percentage for ease of interpretation

**Methods:**

Study population: define adults

Sampling: explain how simple random sampling was carried out

Data collection: who collected the data, who conducted the interviews, which variables were collected through data abstraction, and which variables were collected through face-face interview?

Data analysis: please present the section in logical order. The initial step will be data exportation to Stata and then analysis. Also explain whether the data was initially exported from EMR in MS Excel and then imported to Stata for further analysis? How about variables collected via face-face interview, were they electronically collected, manually in MS Excel e.t.c. please be explicit to ease reproducibility.

Results:

Line 169: majority aged 0-30; however, inclusion criterion says 18yrs and above.

Table 2: label column 2. N?

**Discussion:**

Line 226-227 stated “In this study, we found that PLHIV receiving care in community models had higher rates of return to treatment as compared to those in the facility models.” However, results section Lines 187-188 stated that “Clients in the community care model had a lower return rate (83.9%) compared to those in the facility model (57.4%)”.

**Acknowledgement:**

Please acknowledge study participants.

We look forward to receiving your revised manuscript.

Kind regards,

Jahun Ibrahim, MD, MSC, PhD

Academic Editor

PLOS ONE

**Journal Requirements:**

1. When submitting your revision, we need you to address these additional requirements. Please ensure that your manuscript meets PLOS ONE's style requirements, including those for file naming. The PLOS ONE style templates can be found at https://journals.plos.org/plosone/s/file?id=wjVg/PLOSOne_formatting_sample_main_body.pdf and https://journals.plos.org/plosone/s/file?id=ba62/PLOSOne_formatting_sample_title_authors_affiliations.pdf 2. PLOS requires an ORCID iD for the corresponding author in Editorial Manager on papers submitted after December 6th, 2016. Please ensure that you have an ORCID iD and that it is validated in Editorial Manager. To do this, go to ‘Update my Information’ (in the upper left-hand corner of the main menu), and click on the Fetch/Validate link next to the ORCID field. This will take you to the ORCID site and allow you to create a new iD or authenticate a pre-existing iD in Editorial Manager. 3. Please amend either the title on the online submission form (via Edit Submission) or the title in the manuscript so that they are identical. 4. Please include captions for your Supporting Information files at the end of your manuscript, and update any in-text citations to match accordingly. Please see our Supporting Information guidelines for more information: http://journals.plos.org/plosone/s/supporting-information. 5. If the reviewer comments include a recommendation to cite specific previously published works, please review and evaluate these publications to determine whether they are relevant and should be cited. There is no requirement to cite these works unless the editor has indicated otherwise. 

Reviewers' comments:

**Comments to the Author**

1. Is the manuscript technically sound, and do the data support the conclusions?

Reviewer #1: Yes

Reviewer #2: Yes

2. Has the statistical analysis been performed appropriately and rigorously?

Reviewer #1: No

Reviewer #2: Yes

3. Have the authors made all data underlying the findings in their manuscript fully available?

Reviewer #1: No

Reviewer #2: Yes

4. Is the manuscript presented in an intelligible fashion and written in standard English?

Reviewer #1: Yes

Reviewer #2: Yes

**Reviewer #1:**  Dear Editor,

Thank you for the opportunity to review the manuscript "Factors Associated with Return to Treatment Following Interruption Among HIV-Positive Clients on ART in Public Health Facilities in Katakwi District, North Eastern Uganda".

The study is timely and relevant, addressing the critical issue of treatment interruption and return to HIV care, particularly in rural Uganda—a context often underrepresented in research. Its mixed-methods approach, combining structured quantitative data with contextual insights, enhances understanding of the behavioral and structural factors influencing return to treatment. By grounding the analysis in local realities such as community beliefs, proximity to health facilities, and family support, the findings offer practical value for program design. Furthermore, the use of routine data from the OpenMRS system, triangulated with paper-based records, strengthens the validity and reliability of the results.

However, there is room for improvement. Below are suggestions to help the authors to improve on the current draft and specific recommendations.

General comments

Language and grammar

The manuscript would benefit from language polishing:

• Typo: “client workers” instead of “clients” – Line 165

• Some long sentences, e.g.: “Clients coming from communities with strong beliefs against missing ART or treatment were about three times more likely...” could be re-worded, e.g. “Clients from communities with strong beliefs discouraging treatment interruption were nearly three times more likely to return to care than those from communities without such beliefs—consistent with findings from the United States and Tanzania.” It is not clear how the “strong believe” was measured or how these communities are organized into distinct groups with/without strong beliefs.

Formatting

• Please ensure that the intext citations are properly formatted, e.g., 42 and 30 in line 244 should appear as (42, 30) and not (42)(30). Also check similarly wrongly placed citations, e.g (35)(18) in line 235.

• Some sentences start without a preceding full stop, e.g. the sentence that starts with the words “Facility and home-”

Title

The title is lengthy and will benefit from a revision. For example, a shorter form could be “Determinants of Return to HIV Treatment After Interruption in Katakwi District, Uganda”

Abstract

For better flow, the sentence that starts with the words “Despite these efforts...” needs some modifications to flow better. It is unclear which efforts are referred to.

Introduction

• The introduction is comprehensive but slightly repetitive in describing global/regional return-to-treatment statistics.

• Consider clearly stating the research gap earlier in the introduction.

Methods

• Study design: Though the study follows STROBE, it claims to be mixed-method but only presents structured quantitative data.

o Recommendation: Remove “mixed-method” or clarify any qualitative component.

• Sampling and Exclusion Criteria: The rationale for excluding psychiatric patients and those admitted is not well explained.

• Measurement of outcome (Return to Treatment): Defined as >28 days, which aligns with MOH guidance —but consider justifying the cutoff with references.

• Variable Definitions: Several predictors are mentioned with yes/no responses—consider grouping by domain in a table.

• It may have been helpful to include more facilities with varying levels, since patients may not return to referral facilities since they may prefer to seek care in facilities closest to them. This can be admitted as a limitation.

• It is not clear why the two hospitals were selected as well and also, they seem to be at the same level.

• The sentence “Return to treatment was assessed using frequency counts, percentages, and a 95% confidence interval” can be dropped or re-worded since this is not have been the tool used to assess the outcome variable. Consider describing the measures used for comparison of the proportions in the bivariate analysis.

Results

• Bivariate Table (Table 2): The formatting is inconsistent. For instance, the column “Yes (n=)” is incorrectly formatted. No need to repeat “...of respondent” for the variable Gender. The table title is also long and there is not point stating that the table is about “Bivariate analysis...”

• Please be consistent with the number of decimal places for the p-values. Since the columns figures in brackets are percentages, the symbol can be included in the column headings and that it doesn’t have to be repeated.

• Highlighting of the significant values should be consistent. For example, “Mode of care”. Consider ordering categories for ordinal variables. For example, “Distance to clinic” should start with <5kms.

• Multivariate Table (Table 3): Should be better structured. Include sample size and N per group. Some p-values are misaligned.

Discussion

• Though it has a strong alignment with literature, some considerations on overstating the findings, e.g. in statements like “clients closer to facilities are less likely to return” require cautious interpretation. There may be possible confounding (e.g., stigma, facility congestion) should be hypothesized explicitly. It is surprising that clients who lived <5 kms to the clinic had the highest proportion of IIT, hence it may be challenging to explain why the significance without secondary analysis to rule out or consider stigma as a reason. Stigma alone may also not be the attributable factor since the type of facility and the services may differ depending on the type of facility. Clustering analysis may have been helpful.

• The first sentence in the discussion needs rephrasing since use of the word “only” while referring to 63.9% is countered by the second sentence where the authors indicate that the rate of return is higher than other studies. The rate seems to be within the literature quoted in the second sentence.

• In a vast rural setting, it is possible to contextualize access since some areas may be better connected than others. Hence the sentence “However, the geographical contribution to return to treatment cannot be fully contextualized within this study, as the study sites are in rural settings” may need further thought/reworking.

• It would be helpful to add more explanation on why unemployed clients or those with certain beliefs are more or less likely to return.

• In the limitations section, please remove the word “only” in line 259. In the limitation sentence “...we cannot establish a factor preceded or resulted from return to treatment,” needs substantiation since it is possible to do a competing risk analysis. Additionally, some outcomes such as viral suppression could have been considered since there is a temporal relationship with IIT.

Conclusion

• Thought practical and actionable, it could be more focused. Several recommendations are listed; grouping into themes (e.g., economic support, service delivery flexibility) would enhance clarity.

• The sentence “We noted a high return to care rate that still falls short of the UNAIDS 95% target for return to care” needs to be revised since the commonly known 95-95-95 UNAIDS targets do not reference return to care. Line 261-62

References

Some references are inconsistent, e.g. “11. Country P, Plan O. the People’s. 2020;

12. MOH. The 2019 HIV Epidemiological Surveillance Report for Uganda. file///C/Users/MARTIN%20ODOCHI/Downloads/2019-HIV-Epidemiological- Surveillance report-For print.pdf T 2019 HIV Accessed, 19/07/2021. 2019;(March 2020).”

**Reviewer #2:** A good write up and wonderful insight. however, to include analysis on age and sex if available regression analysis

need to also include inclusion and exclusion criteria. Can also include other form of graphic apart from tables use only.

**Do you want your identity to be public for this peer review?** For information about this choice, including consent withdrawal, please see our Privacy Policy

Reviewer #1: No

Reviewer #2: No

---

## [Author Response · Author response to Decision Letter 1]

16 Oct 2025

Subject: Response to Reviewer’s comments

We appreciate your valuable feedback. We have addressed all the comments, and the changes are tracked. Additionally, we have demonstrated in the response letter how each comment was specifically addressed, including the corresponding line and page numbers. We look forward to your favourable editorial decision.

This is an important study that investigated factors associated with return to care after treatment interruption. Findings will certainly help in designing client-centered strategies that will ensure retention on ART, a requirement for VL suppression and reduction in infection transmission. The study, however, has major gaps as illustrated below, in addition to reviewer 1.

Please address these comments, including all the comments raised by reviewers 1 and 2.

Comment: Ethical statement: “Ethical approval and consent to participate. The research protocol was submitted to Lira University's Faculty of Public Health and the Lira University Research and Ethics Committee for approval (LUREC-2024-235).

The statement is unclear whether the study has been approved by the ethics committee or otherwise. Please be explicit about this.

Response: This has been made clear that the study received Ethical approval from Lira University Research and Ethical Committee (LUREC-2024-235), see lines 152-154, page 7.

Introduction:

Comment: Line 54: In Uganda, out of an estimated 1,492,742 people living with HIV, 1,255,975 are on ARVs (11). Comment: Please provide a percentage for ease of interpretation

Response;

This has been changed to In Uganda, 80.9% of adults living with HIV were aware of their HIV status, and of these, 96.1% were on ART, see lines 50-57.

Methods:

Comment: Study population: define adult

Response: The word adult has been replaced with “aged 18 years and above,” see line 78.

Sampling: explain how simple random sampling was carried out

Response: This has been corrected. Actually, we used a consecutive sampling technique, whereby all eligible clients from this list with appointments within the study period were interviewed, see lines 108-113, page 5

Comment: Data collection: who collected the data, who conducted the interviews, which variables were collected through data abstraction, and which variables were collected through face-to-face interviews?

Response: Data was collected by trained research assistants, who collected data on clients who had missed appointments and returned to care. Then these clients were interviewed. This has been aligned, see lines 134-139, page 6.

Response: The variables collected using face-to-face interviews have been highlighted, see independent variables section, lines 125 to 133, page 6

Comment: Data analysis: Please present the section in logical order. The initial step will be data exportation to Stata and then analysis. Also, explain whether the data was initially exported from EMR in MS Excel and then imported to Stata for further analysis.

How about variables collected via face-to-face interview, were they electronically collected, manually in MS Excel, etc.? Please be explicit to ease reproducibility.

Response: Thanks, this has been done as guided above, see lines 144 to 147, page 6.

Results:

Comment: Line 169: majority aged 0-30; however, the inclusion criterion says 18 years and above.

Response: This was a typo that has been corrected in the table from 0-30 to 18 to 30, which is the study population.

Comment: Table 2: label column 2. N?

Response: Thanks, Column 2 has been labelled N=335 as guided.

Discussion:

Comment: Line 226-227 stated “In this study, we found that PLHIV receiving care in community models had higher rates of return to treatment as compared to those in the facility models.” However, the results section, Lines 187-188, stated that “Clients in the community care model had a lower return rate (83.9%) compared to those in the facility model (57.4%)”.

Response: Thanks. This was an oversight that has been corrected in Lines 187-188 from “lower return rate” to “higher return rate” for clients in the community model.

Acknowledgement:

Comment: Please acknowledge study participants.

Response: This has been done, and study participants acknowledged, see line 299-300, page 17 under acknowledgment

Journal Requirements:

2. PLOS requires an ORCID ID for the corresponding author in Editorial Manager on papers submitted after December 6th, 2016. Please ensure that you have an ORCID ID and that it is validated in Editorial Manager. To do this, go to ‘Update my Information’ (in the upper left-hand corner of the main menu), and click on the Fetch/Validate link next to the ORCID field. This will take you to the ORCID site and allow you to create a new ID or authenticate a pre-existing ID in Editorial Manager.

Response: This has been done.

Response: This has been done.

Response: This has been done.

Reviewers' comments:

Reviewer's Responses to Questions

Comments to the Author

1. Is the manuscript technically sound, and do the data support the conclusions?

Reviewer #1: Yes

Reviewer #2: Yes

2. Has the statistical analysis been performed appropriately and rigorously?

Reviewer #1: No

Reviewer #2: Yes

Response Reviewer 1: Thanks, the analysis has been repeated using Poisson generalized estimating equations (GEE) with robust standard errors, accounting for clustering at the health facility level and adjusting for variables with p-value <0.2 and clinically relevant variables identified in the literature. The 0.2 was selected to avoid excluding variables like age, sex, and community support that are key in determining return to care. See table

3. Have the authors made all data underlying the findings in their manuscript fully available?

The PLOS Data policy requires authors to make all data underlying the findings described in their manuscript fully available without restriction, with rare exceptions (please refer to the Data Availability Statement in the manuscript PDF file). The data should be provided as part of the manuscript or its supporting information, or deposited in a public repository. For example, in addition to summary statistics, the data points behind means, medians, and variance measures should be available. If there are restrictions on publicly sharing data—e.g., participant privacy or use of data from a third party—those must be specified.

Reviewer #1: No

Reviewer #2: Yes

Response to Reviewer 1. The Data has been attached as supplementary material.

4. Is the manuscript presented in an intelligible fashion and written in standard English?

Reviewer #1: Yes

Reviewer #2: Yes

Response: Thanks, Review 1 and 2.

5. Review Comments to the Author

Please use the space provided to explain your answers to the questions above. You may also include additional comments for the author, including concerns about dual publication, research ethics, or publication ethics. (Please upload your review as an attachment if it exceeds 20,000 characters.)

Reviewer #1:

General comments

Language and grammar

Comment: The manuscript would benefit from language polishing:

• Typo: “client workers” instead of “clients” – Line 165

Response: Thanks, “workers” has been deleted see line 162 page 7

Comment; Some long sentences, e.g.: “Clients coming from communities with strong beliefs against missing ART or treatment were about three times more likely...” could be re-worded, e.g. “Clients from communities with strong beliefs discouraging treatment interruption were nearly three times more likely to return to care than those from communities without such beliefs—consistent with findings from the United States and Tanzania.” It is not clear how the “strong believe” was measured or how these communities are organized into distinct groups with/without strong beliefs.

Formatting

Response: Thanks for the guidance given to reword the above statement. Please note your suggestion has been adopted and the statement rewritten as stated by you, see lines 228-230, page 14

Comment: Please ensure that the in-text citations are properly formatted, e.g., 42 and 30 in line 244 should appear as (42, 30) and not (42)(30). Also check similarly wrongly placed citations, e.g, (35)(18) in line 235.

Response: This has been corrected, see line 224 and 227 page 14

Response;

Comment: Some sentences start without a preceding full stop, e.g., the sentence that starts with the words “Facility and home-”

Response: This has been corrected, and all sentences checked for correct punctuation.

Comment;

Title

The title is lengthy and will benefit from a revision. For example, a shorter form could be “Determinants of Return to HIV Treatment After Interruption in Katakwi District, Uganda”

Response: The Title has been modified as guided to Factors Associated with Return to Treatment Following Interruption Among HIV-Positive Clients on ART in Public Health Facilities in Katakwi District, Uganda.

Abstract

Comment: For better flow, the sentence that starts with the words “Despite these efforts...” needs some modifications to flow better.

Response: The statement has been rewritten as “Returning to treatment following interruptions is vital for maintaining optimal HIV care., Uganda faces a 20% treatment interruption rate, with only 58% of clients resuming treatment. However, evidence on factors associated with returning to treatment remains limited” see line 14-16 page 1

It is unclear which efforts are referred to.

Response: The statement effort has been deleted, and the entire paragraph rewritten

Comment; Introduction

• The introduction is comprehensive but slightly repetitive in describing global/regional return-to-treatment statistics.

• Consider clearly stating the research gap earlier in the introduction.

Response: Thanks, this has been corrected as guided

Methods

Comment: Study design: Though the study follows STROBE, it claims to be mixed-method but only presents structured quantitative data.

o Recommendation: Remove “mixed-method” or clarify any qualitative component.

Response: Thanks, this was an oversight. The mixed method statement has been deleted in the abstract on line 19, page 1

Comment: Sampling and Exclusion Criteria: The rationale for excluding psychiatric patients and those admitted is not well explained.

Response: For the psychiatric Patients, they were excluded due to their inability to provide informed consent, see line 99, page 4.

Comment: Measurement of outcome (Return to Treatment): Defined as >28 days, which aligns with MOH guidance —but consider justifying the cutoff with references.

Response: The Statement has been justified with reference to the Ministry of Health guidelines and PEPFAR guidelines, see line 117 - 119, page 5, and references attached

Comment: Variable Definitions: Several predictors are mentioned with yes/no responses—consider grouping by domain in a table.

This has been done, and the section of independent Variables rewritten, and the repetitive statement of Yes/No has been removed. Please see the section on independent Variables, lines 128-136, page 6

Comment: It may have been helpful to include more facilities with varying levels, since patients may not return to referral facilities, and they may prefer to seek care in facilities closest to them. This can be admitted as a limitation.

Response: This has been added as a limitation. Thanks for the guidance. See line 239, 240 page 15

Comment: It is not clear why the two hospitals were selected as well, and also, they seem to be at the same level.

Response: The reason for selecting the facilities has been stated clearly as “These facilities were selected purposively for being facilities with the most clients in care in Katakwi district” and they are of the same level: two are HC III and 1 is the general Hospital, see line 90 page 4

Comment: The sentence “Return to treatment was assessed using frequency counts, percentages, and a 95% confidence interval” can be dropped or reworded since this was not the tool used to assess the outcome variable. Consider describing the measures used for comparison of the proportions in the bivariate analysis.

Response: Thanks for the guidance, this has been re-worded

Results

Comment: Bivariate Table (Table 2): The formatting is inconsistent. For instance, the column “Yes (n=)” is incorrectly formatted.

Response: This has been corrected and aligned. Please see Table 2

Comment: No need to repeat “...of respondent” for the variable Gender. The table title is also long, and there is no point in stating that the table is about “Bivariate analysis...”

Response: Thanks, this has been corrected, word “Respondent” has been deleted from the gender. Please see Table 2

Comment: Please be consistent with the number of decimal places for the p-values. Since the column figures in brackets are percentages, the symbol can be included in the column headings, and it doesn’t have to be repeated.

Response: We have only used 1 decimal point, and the repeated % sign was deleted in both Table 1 and Table 2.

Comment: Highlighting of the significant values should be consistent. For example, “Mode of care”.

Response. This has been done for Significant values only; the p-value is bold with * across Tables 2 and 3.

Comment: Consider ordering categories for ordinal variables. For example, “Distance to clinic” should start with <5kms.

Response. This has been done and the order of categories aligned, starting with < 5kms, then 5 to 10 km, and then >10 km, see tables 1 and 2

Comment: Multivariate Table (Table 3): Should be better structured. Include sample size and N per group. Some p-values are misaligned.

Discussion

• Though it has a strong alignment with literature, some considerations on overstating the findings, e.g., in statements like “clients closer to facilities are less likely to return,” require cautious interpretation. There may be possible confounders (e.g., stigma, facility congestion) that should be hypothesized explicitly. It is surprising that clients who lived <5 km from the clinic had the highest proportion of IIT, hence it may be challenging to explain why the significance without secondary analysis to rule out or consider stigma as a reason. Stigma alone may also not be the attributable factor since the type of facility and the services may differ depending on the t

---

## [Decision Letter · Decision Letter 1]

11 Nov 2025

Determinants of Return to HIV Treatment After Interruption on ART  Among HIV Positive Clients in Katakwi District, Uganda.

PONE-D-25-25853R1

Dear Dr. Okello,

We’re pleased to inform you that your manuscript has been judged scientifically suitable for publication and will be formally accepted for publication once it meets all outstanding technical requirements.

Kind regards,

Ibrahim Jahun, MD, MSC, PhD

Academic Editor

PLOS ONE

Additional Editor Comments (optional):

Reviewers' comments:

Reviewer's Responses to Questions

**Comments to the Author**

Reviewer #2: All comments have been addressed

2. Is the manuscript technically sound, and do the data support the conclusions?

Reviewer #2: Yes

3. Has the statistical analysis been performed appropriately and rigorously?

Reviewer #2: Yes

4. Have the authors made all data underlying the findings in their manuscript fully available?

Reviewer #2: Yes

5. Is the manuscript presented in an intelligible fashion and written in standard English?

Reviewer #2: Yes

Reviewer #2: The author has responded to all comments and the paper is well written and analysed. He is recommended

**Do you want your identity to be public for this peer review?** For information about this choice, including consent withdrawal, please see our Privacy Policy

Reviewer #2: **Yes:** Ismail Lawal

---

## [Editor Report · Acceptance letter]

PONE-D-25-25853R1

PLOS One

Dear Dr. Okello,

I'm pleased to inform you that your manuscript has been deemed suitable for publication in PLOS One. Congratulations! Your manuscript is now being handed over to our production team.

Kind regards,

on behalf of

Dr. Ibrahim Jahun

Academic Editor

PLOS One